# Emerging Role of GCN1 in Disease and Homeostasis

**DOI:** 10.3390/ijms25052998

**Published:** 2024-03-05

**Authors:** Yota Tatara, Shuya Kasai, Daichi Kokubu, Tadayuki Tsujita, Junsei Mimura, Ken Itoh

**Affiliations:** 1Department of Stress Response Science, Biomedical Research Center, Hirosaki University Graduate School of Medicine, 5 Zaifu-cho, Hirosaki 036-8562, Aomori, Japan; 2Diet and Well-Being Research Institute, KAGOME, Co., Ltd., 17 Nishitomiyama, Nasushiobara 329-2762, Tochigi, Japan; 3Department of Vegetable Life Science, Hirosaki University Graduate School of Medicine, 5 Zaifu-cho, Hirosaki 036-8562, Aomori, Japan; 4Laboratory of Biochemistry, Department of Applied Biochemistry and Food Science, Faculty of Agriculture, Saga University, 1 Honjo-machi, Saga City 840-8502, Saga, Japan; tada@cc.saga-u.ac.jp

**Keywords:** GCN1, amino acid starvation, ribosome, disome, GCN2, RWD domain, ribosomal stress surveillance

## Abstract

GCN1 is recognized as a factor that is essential for the activation of GCN2, which is a sensor of amino acid starvation. This function is evolutionarily conserved from yeast to higher eukaryotes. However, recent studies have revealed non-canonical functions of GCN1 that are independent of GCN2, such as its participation in cell proliferation, apoptosis, and the immune response, beyond the borders of species. Although it is known that GCN1 and GCN2 interact with ribosomes to accomplish amino acid starvation sensing, recent studies have reported that GCN1 binds to disomes (i.e., ribosomes that collide each other), thereby regulating both the co-translational quality control and stress response. We propose that GCN1 regulates ribosome-mediated signaling by dynamically changing its partners among RWD domain-possessing proteins via unknown mechanisms. We recently demonstrated that GCN1 is essential for cell proliferation and whole-body energy regulation in mice. However, the manner in which ribosome-initiated signaling via GCN1 is related to various physiological functions warrants clarification. GCN1-mediated mechanisms and its interaction with other quality control and stress response signals should be important for proteostasis during aging and neurodegenerative diseases, and may be targeted for drug development.

## 1. The GCN1–GCN2 Pathway Regulates the Amino Acid Starvation Branch of the Integrated Stress Response

A wide range of stresses induce translational regulation through the phosphorylation of the eukaryotic initiation factor 2 alpha (elF2α) at Ser51, to regulate the cytoprotective response, which is termed the integrated stress response (ISR) [1,2]. In mammals, four elF2α kinases, i.e., General control nonderepressible 2 (GCN2), eukaryotic translation initiation factor 2-alpha kinase 3 (PERK), interferon-induced, double-stranded RNA-activated protein kinase (PKR), and eukaryotic translation initiation factor 2-alpha kinase 1 (HRI), phosphorylate elF2α in response to amino acid starvation (AAS), ER stress, viral infection, and heme deficiency, respectively [3] (Figure 1). Under these stress conditions, phosphorylated eIF2α reduces the amount of GTP-bound eIF2 by competitively binding to eIF2B, and attenuates the translational initiation in general [4]. Conversely, elF2α phosphorylation increases the translation of specific mRNAs carrying an upstream open reading frame in the 5′-untranslated region, such as the mRNAs for the transcription factors cyclic AMP-dependent transcription factor ATF-4 and CCAAT/Enhancer-binding protein homologous protein (CHOP). ATF4 regulates various target genes related to amino acid synthesis, amino acid transport, apoptosis, and autophagy [4,5]. CHOP expression is regulated by ATF4-mediated transcriptional upregulation, posttranscriptional processing, and modulation at the translational level by eIF2α phosphorylation; moreover, it is involved in apoptotic cell death under severe or prolonged stress conditions [6,7,8]. Extensive studies performed in yeast have clarified that the binding of uncharged tRNAs to the histidyl-tRNA synthetase (HisRS)-like domain of GCN2 leads to its activation in AAS conditions. According to the proposed hypothesis, ribosome-bound GCN1 is required for the transfer of uncharged tRNAs from the ribosomal A-site to GCN2, to activate its kinase activity [1]. Budding yeast cells express GCN1, GCN2, and the ATF4 ortholog GCN4, but not other eIF2α kinases; furthermore, GCN2 is activated in response to various stressors, including glucose starvation, oxidative stress, ER stress, osmotic stress [1], and UV radiation [9], in a GCN1-dependent manner. In mammals, the GCN1–GCN2 pathway is activated not only by AAS [3,10], but also by glucose starvation [11], UV irradiation [12,13], and translational inhibition (see Section 5).

## 2. Molecular Structure of GCN1 and Its Role in ISR Activation

GCN1 is a large cytoplasmic protein that is widely conserved in eukaryotes. GCN1 has been predicted to encompass HEAT repeats throughout its length in yeast [1,14] and in mammals, as assessed by structure modeling using AlphaFold (e.g., https://alphafold.ebi.ac.uk/entry/Q92616 for human GCN1 accessed on 28 October 2022) [15,16]. In fact, the structure of yeast GCN1 was recently confirmed by cryoelectron microscopy (see Section 5) [17]. GCN2 possesses an RWD domain in its N terminus that binds to the C-terminal region of GCN1, namely the RWD-binding domain (RWDBD) [1,18,19] (Figure 2). The analysis performed in yeast demonstrated that the ribosome binding region spans 3/4 of GCN1 from the N terminus, of which 12 basic amino acids at positions 754–796 are responsible for ribosome binding, with their substitution abrogating binding to the ribosome [20]. In its central region, GCN1 possesses a domain that is homologous to the N-terminal HEAT repeat domain of the fungal translation elongation factor 3 (eEF3 in eukaryotes), which binds to GCN20 and mediates GCN2 activation in response to AAS [21,22]. GCN20 has two ATP-binding cassettes (ABCs) in its C terminus, similar to eEF3. Although GCN20 enhances the AAS-induced elF2α phosphorylation, which is not absolutely required for this phenomenon [23]. GCN20 itself can interact with ribosomes, with the interaction being augmented in the presence of ATP; however, the N-terminal region of GCN20, but not the ABCs, is sufficient for binding to GCN1 and activating GCN2 [22]. As a GCN20 ortholog in humans, ABC50/ATP-binding cassette sub-family F member 1 (ABCF1) was identified but failed to replace the function of GCN20 in yeast [24]. Subsequently, another member the ABCF subfamily, ABCF-3, from *Caenorhabditis elegans*, was found in the GCN-1 complex and was involved in the developmental role of GCN-1 (as discussed in the next section); moreover, yeast GCN20 can replace ABCF-3 in *C. elegans* [25]. Because mammalian ABCF3 has a higher similarity to GCN20 than the remaining members of the subfamily do, i.e., ABCF1 and ATP-binding cassette sub-family F member 2 (ABCF2) [26], ABCF3 is also required as a GCN20 ortholog for AAS-induced GCN2 activation.

## 3. Emerging Role of GCN1 through GCN2-Independent Pathways

### 3.1. Identification of an Alternative Amino Acid Deprivation Response Distinct from the Canonical GCN2 Pathway in Mammalian Cells

In mammals, mechanistic target of rapamycin (mTOR) and GCN2 mainly sense quantitative changes in amino acids in cells. In response to AAS, both bacteria [27] and eukaryotes sense amino acid scarcity via uncharged tRNAs. Halofuginone, which is an inhibitor of glutamyl-prolyl tRNA synthetase, increases uncharged tRNAs but does not affect cellular amino acid concentration; thus, halofuginone can induce amino acid deprivation signals without changing the mTOR pathway activity. Kim Y. et al. demonstrated that halofuginone downregulates a set of inflammatory and tissue-remodeling responses in a GCN2-independent manner, such as TNFα-induced mediated MMP13 in human fibroblast-like synoviocytes and TGF-β-induced collagenase A1 gene induction in dermal fibroblasts [28]. In the case of TNFα signaling, halofuginone does not affect *PTGS2*, *IL6*, and *CXCL8*, implying that the halofuginone suppression of TNFα signaling is not simply caused by the inhibition of NF-κB. Curiously, they showed that the suppression of TNFα-induced MMP13 and CXCL10 expression by halofuginone is GCN1-dependent, but not GCN2. Halofuginone also downregulates T_H_17 differentiation and function [29]. Those authors reported that halofuginone inhibits T_H_17 differentiation and the response of T_H_17 to IL-23 in a GCN2-independent manner, although the dependency on GCN1 was not examined.

The deprivation of essential amino acids reactivates the transcription of endogenous retroviruses as well as integrated silenced transgenes, such as plasmids, retroviral vectors, and latent HIV-1 proviruses, in a GCN2-independent manner [30,31]. Although the involvement of GCN1 in endogenous retroviral gene re-activation has not been investigated, the results that ribosome inhibitors induce the response may suggest the involvement of GCN1 [30].

### 3.2. Role of GCN1-Dependent and GCN2-Independent Mechanisms in Mammalian Development

To understand the role of GCN1 in amino acid response and cellular homeostasis, we generated two *Gcn1* mutant mice, i.e., *Gcn1* null mice and *Gcn1* mutant mice that specifically lack the RWDBD (hereafter, termed *Gcn1ΔRWDBD* mice) [10]. *Gcn1* null mice died in the early stage of embryonic development, whereas *Gcn1ΔRWDBD* mice appeared to die soon after birth because of respiratory failure concomitant with the delay of whole-body embryonic growth and development. Consistently, *Gcn1ΔRWDBD* mice were smaller than wild-type mice and the embryonic fibroblasts derived from *Gcn1ΔRWDBD* mice exhibited growth retardation with an increased population of G2/M cells and a decrease in cyclin-dependent kinase 1 (Cdk1) and cyclin B1 expression. Because these phenotypes were not observed in *Gcn2* knockout mice, we surmised that the phenotypes observed in *Gcn1* mutant mice were independent of GCN2. Furthermore, as the GCN1-ΔRWDBD protein expressed mice survived longer than complete GCN1 null mice, GCN1 should play a significant role in embryonic development without association with RWD-possessing proteins (see Section 4).

### 3.3. GCN2-Independent Pathway in Species Other Than Mammals

Recent studies performed in various species have also reported the GCN2-independent roles of GCN1. In *C. elegans*, GCN1 regulates apoptosis. In wild-type animals, all the sister cells of pharyngeal M4 motor neurons died from apoptosis during development, but 12–13% of cells survive in GCN1 mutants [25]. This defect was also observed in ABCF-3 mutants, but not in GCN2 mutants. These apoptosis-promoting effects were observed in most somatic cells during *C. elegans* development in the background of the CED3 mutant, which is a *C. elegans* homolog of caspase. Furthermore, GCN1 and ABCF-3 are required for gonadal cell apoptosis in response to irradiation. In contrast, GCN1 regulates the innate immune response in *Arabidopsis thaliana* in a GCN2-independent, but GCN20-dependent manner [32]. GCN1, but not GCN2, regulates the adaptation to mitochondrial dysfunction and high boron concentration, and the activation of plant immunity against *Pseudomonas syringae* pv tomato (Pst) infection. Oxylipins, which is produced from the oxidation of fatty acids by 9-lipoxygenase (9-LOX) and 13-LOX, regulate the plant stress-defense response and immunity. Interestingly, non-responding oxylipin (*noxy*) mutants carry an impaired 9-LOX signaling pathway, and one of the *noxy7* mutant-related genes encodes GCN1.

## 4. Function and Structure of the GCN1-Interacting RWD-Domain-Containing Proteins

RWD domains have been identified in mouse Gcn2 and histidyl-tRNA synthetase, as well as in other eukaryotic species, and are also present in WD-repeat-containing proteins, yeast DEAD (DEXD)-like helicases, many RING finger-containing proteins, and other hypothetical proteins. The RWD domain is named after proteins containing RING fingers, WD repeat-containing proteins, and DEAD-like helicases. The RWD domain consists of approximately 110 amino acid residues and is conserved from yeasts to vertebrates. The function of the RWD domain has been proposed as a surface of protein–protein interactions and is thought to be related to the ubiquitin-conjugating enzyme domain, although the catalytic cysteine is not conserved in most members of this family [33].

Among the RWD-domain-containing proteins (Table 1), GCN2/EIF2AK4 [1], IMPACT [34,35], DFRP2/RWDD1 (Gir2 in yeast) [36,37], and RNF14 [38] are reported to bind to GCN1. Figure 3 shows the results of docking simulations based on the sequence information of the RWD-binding domain of GCN1 and the RWD domains using AlphaFold2 program [16]. The overall structure of the RWD-binding domain of GCN1 was predicted with high confidence (Figure 3A), and the binding surface around Arg-2312 (Arg-2259 in yeast), which is essential for binding to the RWD domain of GCN2, were deduced (Figure 3B,C). Similarly, the amino acid residues which may be involved in the binding to GCN1 were predicted for other RWD-domain-possessing proteins. These results showed that almost identical sites were involved in the interaction with GCN1 in all binding simulations except for WD repeat-containing protein 59, WDR59 (Figure 3D,E). Thus, it appears that the RWD domain can universally bind to GCN1; however, the conditions required for its binding to GCN1, as observed for the binding of GCN2 to GCN1, may be different in each RWD-domain-containing protein.

### 4.1. GCN2/EIF2AK4

As described in the previous section, GCN2 plays an important role for sensing AAS stress, but also in regulating global translation [39]. The GCN2 protein comprises five domains: an RWD domain acting as a regulatory site, a pseudokinase domain, a kinase domain, a HisRS-like domain that binds uncharged tRNA, and a C-terminal dimerization domain (Figure 2).

Under unstressed conditions, the GCN2 forms an inactive homodimer via the C-terminal dimerization domain, HisRS-like, and kinase domains [40]. Uncharged tRNA is a factor that is required for GCN2 activation by the autophosphorylation of GCN2. Under AAS conditions, the concentration of deacylated tRNA (uncharged tRNA) is increased, since tRNA cannot be aminoacylated by tRNA synthetase in the absence of available amino acid. The interaction between GCN2 and uncharged tRNA is dependent on the HisRS-like domain [41] and C-terminal dimerization domain [41], and there is an inverse correlation between the concentration of deacylated tRNA and GCN2 activity [42]. The AAS-mediated GCN2 activation requires GCN1 and GCN20 in yeast [21,22]. We have previously shown that mammalian GCN1 is also necessary for GCN2 activation in response to AAS and UV irradiation using mouse embryonic fibroblasts harboring a GCN1 mutant lacking the GCN2-binding domain [10].

In terms of physiological function, GCN2 is involved in the regulation of neuronal functions, synaptic plasticity, memory [43], and feeding behavior [44]. In humans, *GCN2* mutation causes heritable pulmonary veno-occlusive disease, a rare subgroup of severe pulmonary arterial hypertension [45]. IMPACT, which is an RWD-domain-possessing protein that is similar to GCN2 (as described below), is upregulated during neuronal differentiation, whereas GCN2 activity is downregulated. IMPACT binds to ribosomes to enhance the translation initiation and downregulate the ATF4 expression. Furthermore, IMPACT promotes neurite outgrowth, whereas GCN2 inhibits spontaneous neuritogenesis [46]. Considering that GCN1 binds to both GCN2 and IMPACT via respective RWD-binding domains, we speculated that GCN1 is involved in neurite outgrowth. Curiously, a low-molecular-weight stabilizer of the 14–3–3 interactome, fusicoccin-A, stabilizes a complex between 14–3–3 and GCN1 and degrades GCN1, leading to axonal outgrowth [47]. The elucidation of the role of GCN1 in neuronal differentiation is eagerly awaited.

### 4.2. IMPACT

Human IMPACT is a protein comprising 320 amino acid residues, with an RWD domain at its N terminus and a domain of unknown function at its C terminus. The fact that yeast IMPACT homolog 1 (Yih1) suppresses GCN2 activity suggests that Yih1 abrogates the interaction between GCN1 and GCN2 in amino-acid-starved cells [34,48,49]. Mouse IMPACT also inhibits the GCN1–GCN2-mediated response in mouse embryonic fibroblasts [49]. Because the suppression of GCN2 activity results in enhanced protein synthesis, IMPACT is a translation regulator that assures consistent protein translation under AAS. Since Yih1 deletion does not increase eIF2α phosphorylation, Yih1 resides in the actin complex under normal conditions, and may be released from the actin complex to inhibit GCN2 under specialized conditions [48]. On the other hand, the knockdown of the IMPACT homolog impt-1 activates the ISR pathway in *C. elegans* and increases the lifespan and dietary restriction stress resistance in a Gcn2-dependent manner [50]. The structural similarity to yeast Yih1 suggests that it plays a similar role in mammals. In mouse neurons, the IMPACT expression is higher compared with other tissues [34,46]; moreover, the suppression of IMPACT inhibits neuritogenesis by increasing the basal levels of GCN2 activation [46].

### 4.3. DFRP2/RWDD1

Human DFRP2 is a 243-amino-acid protein composed of an RWD domain at the N terminus and a binding region to DRG2 at the C terminus. DGRP2 is a conserved binding partner of DRG2 [51] that protects DRG2 from proteolytic degradation via the ubiquitin–proteasome system, although neither the ubiquitin site nor the responsible E3 ubiquitin ligase has been identified [52]. DRG2 is a member of the developmentally regulated GTP-binding protein (DRG) subfamily and is one of two highly conserved paralogs that are thought to be involved in the regulation of cell proliferation, translation, and microtubules [53]. The yeast homologs of DFRP2 and DRG2 are Gir2 and Rbg2, respectively. Gir2 binds to yeast GCN1, and they are cofractionated in the polysome [36]. A recent analysis of *C. elegans* also demonstrated that GCN1 is co-immunoprecipitated with DFRP2 and DRG2 [54].

### 4.4. RNF14 and RNF25

RNF14 is a protein that is composed of 474 amino acid residues, an RWD domain, and a TRIAD supradomain. Because all characterized proteins containing TRIAD supradomains exhibit E3 ligase activity, it is speculated that RNF14 also has E3 ligase activity [55]. RNF25 shares the same molecular size (459 amino acid residues) and RING-type zinc finger domain as RNF14 and appears to possess E3 ligase activity. RNF14 and RNF25 act as translation elongation checkpoints that monitor ribosomal GTPase centers, such as eukaryotic elongation factor-1α (eEF1α).

### 4.5. RESUME/RWDD3

RESUME is a protein with an RWD domain consisting of 267 amino acids; however, no other characteristic domains have been identified. The crystal and solution structure of the RESUME RWD domain was determined to be a binding domain to ubiquitin-conjugating enzyme E2I, yeast UBC9 homolog (UBC9/UBE2I) [56]. RESUME enhances small ubiquitin-related modifier (SUMO) conjugation by promoting thioester linkage between SUMO and UBC9/UBE2I, and transfers SUMO to specific target proteins, including hypoxia-inducible factor 1-alpha (HIF1A), protein inhibitor of activated STAT 1 (PIAS), NF-kappa-B inhibitor alpha (NFKBIA), glucocorticoid receptor NR3C1, and DNA topoisomerase 1 (TOP1); however, it does not affect ubiquitination [57,58,59,60]. The binding of RESUME to GCN1 has not been reported, and it is not known whether GCN1 is involved in SUMO conjugation.

## 5. Emerging Role of GCN1 in Co-Translational Protein Quality Control

Translating ribosomes can slow down or stall under stressed conditions, such as ribotoxin exposure, UV radiation, and AAS, and even under unstressed conditions, such as when ribosome encounters codons which are difficult to be translated. When mRNA is somehow damaged and the ribosomes are stalled, several co-translational quality control mechanisms in the translation machinery are initiated [61,62]. Stalled ribosomes are split from mRNA for recycling, and arrested nascent polypeptide chains are degraded by the ribosome-associated quality control (RQC), whereas mRNA is degraded by nonsense-mediated decay (NMD), no-go decay (NGD), and nonstop decay (NSD) (Figure 4). When the ribosomes themselves are damaged, another surveillance process called nonfunctional ribosomal decay degrades the nonfunctional ribosome (Figure 4).

Recently, a cryoelectron microscopic analysis performed in yeast revealed that GCN1 binds to disomes [17] suggesting that GCN1 is involved in co-translational protein quality control (Figure 5). A cyclic peptide, Ternatin-4, which locks the aminoacyl-tRNA–eEF1α complex at the ribosomal A-site and inhibits translation, causes eEF1α degradation in a translation- and GCN1-dependent manner [38]. The eEF1α catalyzes the GTP-dependent binding of aminoacyl-tRNA to the A-site of ribosomes during protein synthesis [63,64]. GCN1 binds to the RWD domain of RNF14 via RWDBD, and this interaction together with the additional action of E3 ligase RNF25 is essential for eEF1α degradation, suggesting that GCN1 functions to resolve ribosome stalling. However, the functional consequences of the reaction were not examined. Gurzeler L. A. et al. showed that the GCN1–Rnf14 interaction mediates eRF1 degradation when premature termination codon readthroughs are enhanced by chemical readthrough promoters [65]. In turn, readthrough promoters induce ribosome collisions by locking eRF1 at the ribosomal A-site and subsequently induce eRF1 degradation, to induce readthrough. Thus, it appears that GCN1 acts to release ribosome collision when translation is inhibited, either during translation elongation or premature translation termination, with its A-site site occluded by tRNA.

The disome is a structural unit that is commonly recognized by both the RQC factor ZNF598 (homolog of yeast Hel2) [61,66] and the ribosome-initiated stress response factors GCN1 and mitogen-activated protein kinase kinase kinase AZK long isoform (ZAKα). Below, we summarize each of these pathways and discuss their inter-relationships and the specificity of their recognition.

### 5.1. Quality Control Mechanisms for Stalled Ribosomes

In RQC, the disomes formed by stalled and colliding ribosomes trigger the recruitment of the E3 ubiquitin ligase ZNF598 in mammals, which is an ortholog of yeast Hel2, and the subsequent ubiquitination of the 40S ribosomal subunit proteins of disomes [67,68] (Figure 4C). The ZNF598-mediated ubiquitination of 40S ribosomal proteins, i.e., small ribosomal subunit protein eS10 (RPS10) and small ribosomal subunit protein uS10, induces the irreversible dissociation of ribosome subunits by the ASC-1 complex [69] and the subsequent degradation of the mRNA and nascent peptide [70]. A persistent collision can recruit the endothelial differentiation-related factor 1 (EDF1, an ortholog of yeast multiprotein-bridging factor 1 (Mbf1) that is found in the GCN1–disome complex [17]), which then recruits the translational repressors GRB10-interacting GYF protein 2 (GIGYF2) and 4EHP/eukaryotic translation initiation factor 4E type 2 (EIF4E2) to the disome, so that a negative feedback loop prevents further translational initiation in a ZNF598-independent manner [71,72]. Because ZNF598 depletion enriches EDF1 and GIGYF2 on collided ribosomes [72,73], independent RQC pathways can redundantly contribute to the resolution of various translational problems. Intriguingly, GCN2 is activated after the Hel2 (also known as Rqt1 in yeast; ZNF598 in mammals)-mediated RQC is overwhelmed [74]. The authors showed that Hel2 ubiquitination in response to methylmethane sulfate is saturated at approximately 10-fold lower concentrations of methylmethane sulfate compared with eIF2α phosphorylation. This indicates that GCN2 is activated in the late phase of methylmethane sulfate-induced ribosome collision. GCN1 may contribute to translational regulation by cross-talking with RQC. In this context, however, GCN2 is activated before RSR and contribute to the cell survival, whereas RSR leads to cell cycle arrest or apoptosis [39,75].

Muller et al. used mRNA pull-down experiments in *C. elegans* with an mRNA with a stop codon mutation to demonstrate that GCN-1 strongly binds to the stop-codon-mutated mRNA, and that GCN-1 recruits the CCR4/NOT complex to degrade the stalled mRNA [54]. Furthermore, using a selective ribosomal profile, the authors showed that GCN1 binds to the disome at nonoptimal codons that mainly exist in 3′UTRs, transmembrane proteins, and collagens in *C. elegans* and HEK293 cells. This result suggests that GCN1 is especially important when stop codon readthrough occurs because of environmental stresses or aging (see Section 8).

### 5.2. GCN2 Branch of the ISR

Although it has been established using mainly yeast analyses that GCN2 is activated by uncharged tRNAs in response to AAS [76], Ishimura et al. reported that ribosome stalling under translational stress activated mouse Gcn2 in the absence of an increase in uncharged tRNA, suggesting that ribosome collision is another mechanism of GCN2 activation [77]. Consistently, the starvation of leucine or arginine can cause ribosome pausing and collisions, albeit to different extents [78]. Considering that GCN1 is essential for GCN2 activation and may bind at least preferentially to disomes compared with monosomes, GCN2 activation in response to AAS may occur in particular disomes. There is an interesting argument that ISR is activated by translation-inhibitor-mediated ribosome collisions only when the A-site of the leading ribosome is vacant (Figure 5A) [74,79]. Yan et al. demonstrated that tigecycline, but not anisomycin and cycloheximide, activates GCN2. However, there are several exceptions where anisomycin, which inhibits the translational elongation by binding to the peptidyl transferase center and may inhibit translation with the A-site tRNA, also activates ISR [39,75]. However, Wu C. et al. demonstrated that A-site tRNA is unstable, at least in vitro [80]. Nevertheless, these results indicate that the status of A-site occupancy modifies the GCN1-mediated co-translational quality control, where GCN2 is a GCN1 partner when the A-site is vacant and RNF14 is a partner when the A-site is occupied in the collided ribosome. Another interesting point is that GCN1 interacts with RPS10 in an RNA-independent manner, and this interaction is required for the full activation of GCN2 [81]. Because RPS10 ubiquitination is also required for RQC, this may explain both the vacant A-site preference for GCN2 activation and the antagonism of ISR and RQC.

eIF2α phosphorylation induced by UV radiation involves RNA photodamage and ribosome collisions and requires GCN2 [39]. Previously, we demonstrated that this response also requires GCN1; therefore, it appears that GCN1 is important for ribosome collision-mediated GCN2 activation in response to UV irradiation. The increase in ribosome collisions triggered by low-dose anisomycin or UV radiation induces the ZAKα-mediated activation of both the ribotoxic stress response (RSR) and the GCN2–eIF2α axis of ISR [39], although the precise mechanism underlying the activation of GCN2 by ZAKα remains unclear. Because GCN2 activation by nutrient starvation or AAS does not require ZAKα, it has an alternative mechanism for GCN2 activation [82]. Of note, ribosome collision may also activate ISR in response to environmental toxicants, such as 1-nitropyrene [83] and formaldehyde [84], as well as pharmacological agents [84]. These reagents and UV leave the A-site vacant during translation inhibition. Furthermore, the signaling molecule nitric oxide also activates ISR and RSR via ribosome collision (see below) [85]. Thus, GCN1–GCN2 and other eIF2α kinases function as sentinels that monitor translational defects and proteostasis via ISR. Interestingly, the RQC factor ASC-1 complex can split ribosomes stalled by usual elongation inhibitors, such as anisomycin, but not ribosomes stalled by damaged mRNA with a bulky modification of UV light B (UVB) or 4-nitroquinoline 1-oxide [79]. Under these stresses, the sustained or enhanced activation of GCN2 and ZAKα is observed.

The direct activation by ribosomes, particularly ribosome P-stalks, is another uncharged tRNA-independent mechanism of activation of GCN2. Both uncharged tRNA and ribosome P-stalks activate GCN2 in vitro [76]; however, P-stalks [76,77] activate GCN2 more robustly [86]. Because the P-stalk is located near and just above the ribosomal A-site, GCN2, which is properly localized by GCN1 in disomes, may be activated by the P-stalk.

mRNAs that possess the 5′-terminal oligopyrimidine (5′ TOP) motif at the 5′UTR encode ribosomal proteins. The ATF4 target LARP1 recognizes the 5′ TOP motif in mRNA and inhibits the translation of ribosomal proteins and translation factors under nutrient starvation [87]. Interestingly, LARP1 is recruited to the GCN1–disome complex in response to increased ribosome collision [87].

### 5.3. RSR

RSR was identified by the activation of p38 MAPK and JNK in response to the antibiotic anisomycin and enzymes that interfere with a large subunit of the active ribosome [88]. RSR can also be elicited by the translation inhibitor cycloheximide, the antitumor agent doxorubicin, and UV radiation; and induces inflammatory signals as well as apoptotic cell death [89]. p38 MAPK and JNK can be activated by several stimulants, such as growth factors, inflammatory cytokines, and external and intrinsic stress signaling via the upstream kinase cascade of specific MAPKKs and MAPKKKs. As a specific MAPKKK responsible for RSR, Wang X. et al. found that ZAK/MLK7/MAP3K20 mediates the p38 and JNK activation induced by anisomycin and UV radiation [90], and that ZAK is essential for RSR triggered by the Shiga toxin, ricin [91,92], and doxorubicin [93]. The vertebrate *ZAK* gene encodes two alternatively spliced variants that differ at their C termini, with the longer isoform (ZAKα) harboring two ribosome-interacting domains that sense ribosome conformation. ZAKα, but not the short ZAKβ isoform, mediates RSR triggered by anisomycin and UV radiation [39,94]. Mechanistically, ZAKα can sense both ribosome stalling and ribosome collision [82]. GCN1 and ZAKα co-migrate with the RNase-resistant disome fraction in a ZAKα-dependent manner, and ZAKα is also necessary for the ribosome-collision-mediated activation of both RSR and GCN2-mediated ISR, although the manner in which ZAKα participates in GCN2 activation remains unclear. Interestingly, both mRNA methylation by methylmethane sulfate and bulky mRNA modification by genotoxic agent cause ribosome collision [79]. However, disomes formed by bulky adducts, but not by methylation, are resistant to ribosome splitting by the ASC-1 complex. Thus, prolonged collisions by bulky adducts triggered by UV light B (UVB), 4-nitroquinoline, or mitomycin C activate RSR and cause G2/M cell-cycle arrest [79]. Furthermore, ZAKα activates the human Nod-like receptor 1 (NRLP1) [95], which was recently shown to be involved in senescence [96]. Consistently, ZAKα mediates oxidative stress-mediated induction of SAPK activation [97].

## 6. Function of GCN1 in Basal-State Translational Regulation

The rate of elongation of nascent proteins by ribosomes is not constant; rather, it depends not only on the availability of aminoacyl-tRNA, but also on mRNA and nascent proteins [98,99,100,101,102,103,104]. Ribosome speed is high for efficient protein synthesis, and low for assisting regulatory events, such as the folding and localization of nascent proteins, protein–protein interactions, and programmed frameshifts [105]. Ribosomes stall during translation for various reasons, such as the so-called rare codons. The number of tRNAs that recognize codons that do not appear frequently is low; therefore, ribosomes are more likely to stall and cause a translation error at these rare codons. GCN2 not only suppresses the initiation of protein synthesis via the ISR pathway, but also inhibits the translational elongation of genes with rare codons in a CDS-length-dependent manner [106].

It is interesting to note that *DRG2* KO mice exhibit similar defects in embryonic growth with GCN1 mutant mice, although *GCN1ΔRWDBD* mice die soon after birth, whereas *DRG2* KO mice survive into adulthood with a decreased body weight [107]. Thus, DRG2 may cooperate with GCN1 during cell proliferation. DFRP2–DRG2 has been suggested to be involved in translation regulation. Ishikawa et al. showed that Gir2 overexpression dissociates Gcn2 from Gcn1, decreases cell proliferation under reduced amino acid conditions, and increases Gir2–Rbg2 and Gcn1 binding during AAS [37]. Recently, the cryoelectron microscopy structure of yeast Gcn1 in the complex with stalled and colliding 80S ribosomes and the Gir2–Rbg2 complex was resolved [17]. In the structure, the C terminus of Gcn1 is suggested to be in contact with the RWD domain of Gir2, in agreement with previous reports [36,37,51]. Interestingly, the complex also contains MBF1 and eukaryotic translation initiation factor 5A-1 (eIF5A), which are important for the inhibition of the frameshifting [108] and the translational elongation of particularly difficult codons, such as proline-rich peptides [109]. In addition, eIF5A was recently shown to be important for RQC by enhancing CAT-tailing (see below) [110]. The Gir2–Rbg2 complex may promote the translation elongation of difficult codons by binding to the stalled ribosome before Gcn2, as suggested by Zeng F. et al. [111]. Kriachkov V. et al. demonstrated that DRG2 promotes the translation elongation of poly-lysine stretches [112]. Thus, although it remains unknown whether the interaction between yeast Gir2 and Gcn1 is conserved in higher eukaryotes, GCN1 may be important for the translation elongation of several difficult proteins. In contrast, GCN1 may accelerate the translation elongation of rare codons together with DFRP2/DRG2, as discussed above. In yeast, GCN1 inhibits frameshifting at difficult codons [108]. Thus, GCN1 may play a housekeeping role in translation. When stalling is prolonged, GCN1 may switch partner to GCN2, to activate it. Finally, the enhanced activation of p38 in response to RQC-resistant ribosome stalling induced by UVB and 4-nitroquinoline 1-oxide, as discussed above, causes G2 m cell-cycle arrest, which is similarly observed in GCN1 or DRG2 mutant cells [79].

## 7. Role of GCN1-Mediated Responses in Energy Homeostasis

The ISR effector ATF4 is involved in energy homeostasis. ATF4 KO mice are lean and resistant to high-fat diet-induced obesity [113]. Conversely, ATF4 is activated by elevated fat levels in the liver and pancreas [114,115]. The activation of ATF4 in adipose tissue (AT) and brown AT similarly improves obesity and glucose intolerance [116,117]. *GCN2* KO mice are also resistant to the impairment of glucose tolerance in model mice with type 2 diabetes [118], and to hepatic steatosis in *ob*/*ob* mice or after high-fat diet ingestion [119]. The GCN2 pathway, which is responsive to AAS, affects the feeding behavior by activating a neuronal circuit that biases consumption against foods with an imbalanced amino acid content [44], indicating that the GCN1–GCN2 pathway is associated with systemic energy homeostasis. This section will focus on GCN1 in type 2 diabetes mellitus and liver energy metabolism.

### 7.1. Role of the GCN1–GCN2 Pathway in the Regulation of Insulin Secretion and Sensitivity

It is known that SNPs of *Cdkal1* are involved in the etiology of type 2 diabetes [120]. Cdkal1 is important for lysine tRNA modification for accurate lysine decoding and subsequent processing of pro-insulin [121]. Cysteine persulfide has been proposed to mediate the thiolation (2-methylthio-*N*(6)-threonylcarbamoyladenosine (ms(2)t(6)A)) of tRNA(Lys)(UUU) at position 37 [122]. Interestingly, GCN2 SNPs are involved in the decreased insulin secretion and/or insulin sensitivity observed in the Japanese population with healthy and/or impaired insulin tolerance [123]. GCN2 is activated during a high-fat diet [114], when the translation of pro-insulin is highly activated [124]. Furthermore, it was recently shown that ATF4 protects islet cells against ER stress [125]. The activation mechanisms that regulate the activation of GCN2 require further exploration. Interestingly, GCN1 is one of the most downregulated proteins in the pancreatic islets when the insulin gene is genetically halved, suggesting that GCN1 is involved in insulin translation [126].

### 7.2. GCN1 Regulates Energy Storage and Usage

Using mice with floxed alleles, we generated conditional *Gcn1* knockout mice (*Gcn1* CKO mice). After the administration of tamoxifen to *Gcn1* floxed mice with Cre recombinase fused to ERT2, i.e., *Gcn1* CKO mice, the animals exhibited a decrease in body weight concomitant with decreased liver and adipose tissue weight per body weight, although the mice were viable and consumed the same amount of food. Interestingly, these changes in body weight disappeared after the cessation of tamoxifen treatment, and re-appeared upon subsequent tamoxifen treatment, indicating the reversible nature of the phenomena and the involvement of both *Gcn1* deletion and the toxicity of tamoxifen. The liver triglyceride and glycogen content, as well as the blood glucose and non-esterified fatty acid levels, were significantly decreased in tamoxifen-treated *Gcn1* CKO mice compared with the negative control (tamoxifen-treated *Gcn1* floxed mice without Cre recombinase). A liver proteome analysis indicated the increased peroxisomal fatty acid β-oxidation and decreased mitochondrial fatty acid β-oxidation, thus explaining the inefficient energy production that results in food-energy dissipation as H_2_O_2_ (Figure 6) [127]. Interestingly, *ZAKα* KO mice showed the decreased p38 phosphorylation in response to leucine starvation, but also exhibited decreased fat tissues, increased the browning of white adipose tissues, and decreased the liver triglyceride levels at the basal level [82]. Furthermore, the liver of *ZAKα* KO mice displayed a similar increase in *ACOX* gene expression at the basal level, indicating increased peroxisomal β-oxidation, as observed in *Gcn1* CKO mice. Thus, signals from collided and/or stalled ribosomes may somehow regulate energy homeostasis.

## 8. Potential Role of GCN1 in Aging and Disease

### 8.1. Potential Role of GCN1 in Aging

Abnormal proteostasis is one of the 12 hallmarks of aging [128]. Although posttranslational protein homeostasis mechanisms, such as proteasome- and autophagy-mediated protein degradation, have received attention in the context of aging, the functional decline of translational efficiency and fidelity is also an important factor during aging. In fact, it was reported that aging increases ribosome pausing [129] and that stop codon readthrough increases with age, especially in neurons [130]. Muller et al. demonstrated that the translation readthrough of stop codons activates GCN1-dependent eIF2α phosphorylation [54]. The authors further demonstrated that transmembrane protein aggregation is increased in aged GCN1 mutant worms, demonstrating the role of GCN1 in protein homeostasis during aging. Translational termination was mediated by eukaryotic release factor 1 (eRF1), eukaryotic release factor 3 (eRF3), and ABCE1. Interestingly, 3′ readthrough is increased during aging, most probably because ABCE1 is an iron–sulfur protein that is susceptible to lysosomal dysfunction, iron deficiency, and oxidative stress [131]. It was first reported that CAT-tailing occurs in stalled nascent polypeptide chains when the degradation of nascent polypeptide chains by the RQC protein E3 ubiquitin-protein ligase listerin (Ltn1) is inhibited and enhances the degradation of stalled proteins [132]. CAT-tail-like C-terminal extensions were observed in the brains of *Drosophila* PINK1 mutants [133]. A set of nuclear-encoded mitochondrial proteins are co-translationally transported into the mitochondrial outer membrane (MOM); moreover, mitochondrial ROS production, which is increased during aging, enhances the readthrough of the stop codons. PINK1–PARKIN regulates nuclear-encoded respiratory chain complexes that are translated on the MOM [134]. CAT-tail-like C-terminal extensions were enhanced in the mitochondrial complex-I 30 kDa subunit (C-I30) when translational termination was impaired by mitochondrial defects, which is termed mitochondrial stress-induced translational termination impairment and protein carboxy extension, MISTERMINATE [133]. As discussed above, GCN1 is selectively recruited to the 3′UTR upon stop codon readthrough. Thus, GCN1 function may be associated with the aging process.

### 8.2. Potential Role of GCN1 in Neurodegenerative Diseases

Mutations in RQC factors such as Ltn1 [132] and nuclear export mediator factor (NEMF) [135] and ribosomal stalling in the brain triggered by the scarcity of brain-specific tRNAs cause neurodegenerative diseases [136]. As discussed above, stop codon readthrough increases with age, particularly in neurons [130]. Consistently, isolated 3′ UTRs, which may be a consequence of 3′ readthrough and a product of no-go decay mRNA cleavage, accumulate during aging and are associated with mitochondrial dysfunction and oxidative stress [137]. Of note, the role of CAT-tail-like C-terminal extensions in neurodegeneration has been noted in Alzheimer’s disease and amyotrophic lateral sclerosis/frontotemporal dementia [138]. APP.C99, the precursor of β-amyloid, can cause neurodegeneration independently of amyloid β [139]. Ribosomes stall during the co-translational transport of APP.C99 in the ER membrane and Are rescued by RQC; however, insufficient RQC adds CAT-tail-like C-terminal extensions onto APP.C99, which are prone to aggregation and cause lysosomal dysfunction [139]. Interestingly, previously, we demonstrated that miRNAs targeting GCN1 are selectively downregulated in the peripheral blood of patients with amnestic mild cognitive impairment, a pre-disease state of Alzheimer’s disease, indicating that GCN1-mediated quality control may be enhanced in amnestic mild cognitive impairment [140]. In addition, whole-blood transcriptome and plasma metabolome analyses suggested an AAS response in undetermined cells in the peripheral blood. Our data are reminiscent of previous data showing that the mitochondrial stress pathway involving ATF4 (i.e., integrated stress response) is activated during the early phase of Alzheimer’s disease [141]. It is of note that the total GCN1 protein is downregulated, whereas it is increasingly recruited to polysomes during neuronal differentiation [46]. Furthermore, increased interaction with 14–3–3 induces GCN1 degradation and stimulates axon regeneration, indicating a neuronal function for GCN1 [47].

## 9. Future Perspectives

Recently, ISR has been established as the central mechanism of retrograded mitochondria signaling in mammals, which restores mitochondrial function in response to mitochondrial dysfunction. Defects in mitochondrial dysfunction and protein transport in the matrix activate the HRI–eIF2α axis via metalloendopeptidase OMA1-mediated DAP3-binding cell death enhancer 1 (DELE1) processing [142,143] and DELE1 accumulation [144,145], respectively. GCN2 is another mediator of ISR in response to mitochondrial dysfunction [146,147]. Although mitochondrial reactive oxygen species also activate GCN2, the underlying mechanism remains unclear. Mitochondrial defects cause stop codon readthrough and activate the GCN1–GCN2 pathway, as discussed above. Furthermore, Li J. et al. showed that the mitochondrial reactive oxygen species produced by the environmental toxin 1-nitropyrene cause ribosome collision to activate GCN2 in the mouse testis [83]. Thus, ribosome collision may mediate mitochondrial reactive oxygen species-induced GCN2 activation. Mitochondrial reactive oxygen species regulate cytosolic translation by oxidatively modifying ribosomes [148]. GCN1 may be one of the mechanisms of mitochondrial communication, in parallel with HRI-mediated ISR. As discussed in this review, GCN1 acts as a signaling hub and should be involved in many aspects of biology by acting as a central regulator in the final step of gene expression (i.e., translation) by responding to various environmental and endogenous cues. Future studies may shed light on the function of GCN, which is important, but had remained underexplored until recently.

## Figures and Tables

**Figure 1 ijms-25-02998-f001:**
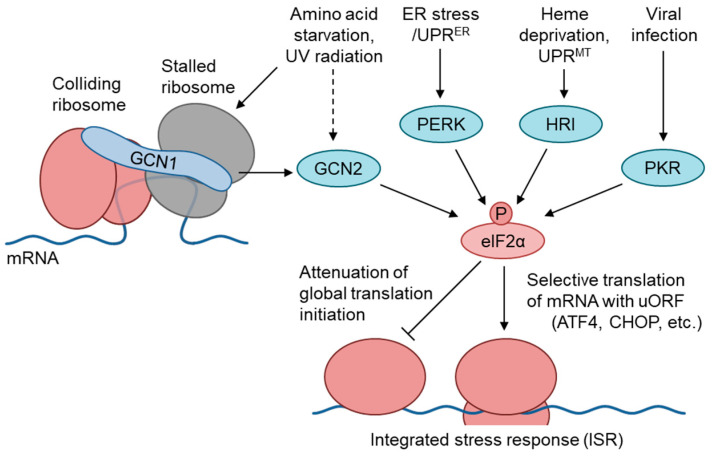
The GCN1–GCN2 pathway mediates the integrated stress response (ISR). In response to various stressors, eIF2α phosphorylation by stress-specific kinases attenuates global translation initiation, which is known as ISR and is conserved in eukaryotes (right part). GCN1 is a scaffold protein that interacts with ribosomes, GCN2, and other RWD-domain-containing proteins, and is essential for GCN2 activation under conditions of amino acid starvation and UV irradiation. Indirect GCN2 activation by these stressors was presented by dashed line. Yeast GCN1 was found in a colliding ribosome (disome) complex under unstressed conditions. The resultant GCN2 and subsequent eIF2α phosphorylation represses cap-dependent translation, but derepresses mRNAs carrying inhibitory upstream open reading frames, such as ATF4 and CHOP, which regulate the transcriptional activation of amino acid metabolism and apoptotic cell death, respectively.

**Figure 2 ijms-25-02998-f002:**
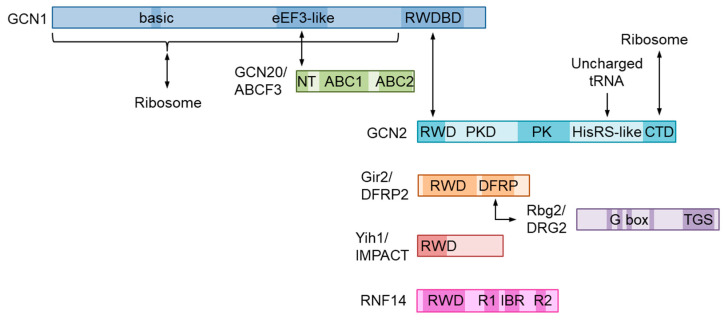
Domain structure of the GCN1- and RWD-domain-containing proteins. GCN1 interacts with ribosomes via its N-terminal 3/4 region three-quarters and with RWD-domain-containing proteins via a C-terminal region named the RWD-binding domain (RWDBD). The GCN1 eEF-like domain interacts with the N-terminal portion of yeast GCN20, which is also conserved in animal ABCF3, with these interactions being essential for AAS-induced GCN2 activation. GCN2 can directly bind ribosomes via its C-terminal domain (CTD) in yeast. Yeast GCN1 and GCN20 form a complex (disome) that includes the RWD-containing protein Gir2 and ribosome-interacting GTPase 2 (Rbg2). RWD domain-containing protein 1 (DFRP2) and developmentally regulated GTP-binding protein 2 (DRG2) form a heterodimer via the DFRP domain and an unknown region of DRG2. IMPACT, which is a protein that is enriched in neuronal cells and is involved in neuritogenesis, and Gir2 can compete with GCN2 for GCN1 because their forced expression inhibits GCN2 activation. Ring finger protein 14 (RNF14), which is an E3 ubiquitin ligase that is involved in the ubiquitination of stalled ribosomes, possesses an RWD domain and two RING domains, RING1 (R1) and RING2 (R2), as well as an in-between RING fingers (IBR) domain.

**Figure 3 ijms-25-02998-f003:**
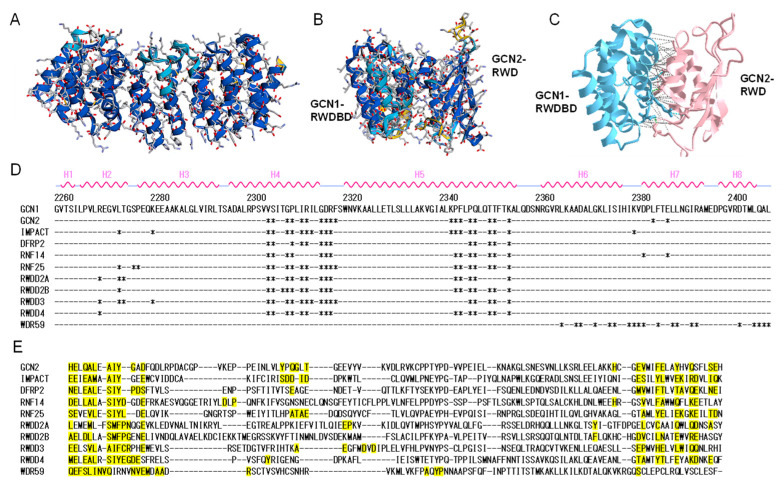
Structure prediction of GCN1 RWDBD and docking simulation using the RWD domain. (**A**) The three-dimensional structure of the RWDBD domain of human GCN1 was constructed according to its amino acid sequence (2260–2408) using the AlphaFold2 program. The confidence of the predicted model is colored in blue, cyan, and yellow for very-high (pLDDT > 90), confident (90 > pLDDT > 70), and low (70 > pLDDT > 50) values, respectively. The model did not include very-low confidence values (pLDDT < 50). (**B**) A docking simulation of human GCN1 RWDBD using the RWD domain of GCN2 was performed using the AlphaFold2 program. (**C**) Interaction analysis of human GCN1 RWDBD with the RWD domain of GCN2. Hydrogen bonds (3.8 Å), salt bridges (6 Å), and π-cation bonds (6 Å) resulting from the interaction analysis using iCn3D (https://www.ncbi.nlm.nih.gov/Structure/icn3d/full.html accessed on 28 October 2022) are denoted by gray, green, and red dashed lines, respectively. The side chains of Arg-2312 of GCN1 and Asp-37 of GCN2, which were involved in a salt bridge, are depicted using stick models. (**D**) RWDBD sequence of human GCN1 and binding sites to RWD domains predicted by docking simulations and interaction analysis. The amino acids in the GCN1 RWDBD that were predicted to be involved in binding to the each RWD domain protein are indicated by asterisks. The α-helix structures H1 through H8 in the RWDBD model are presented at the top of the sequence. (**E**) Multiple alignments of RWD domains. Amino acid sequences of human RWD-containing proteins, GCN2/EIF2AK4 (amino acids 25–137), IMPACT (14–116), RNF14 (11–137), E3 ubiquitin-protein ligase RNF25 (18–128), DFRP2/RWDD1 (10–114), RWD domain-containing protein 2A (RWDD2A) (14–134), RWD domain-containing protein 2B (RWDD2B) (41–165), RWD domain-containing protein 3 (RWDD3) (7–114), RWD domain-containing protein 4 (RWDD4) (9–111), and WDR59 (393–494) were used. The binding sites that were predicted to be involved in binding to GCN1 RWDBD are highlighted in yellow.

**Figure 4 ijms-25-02998-f004:**
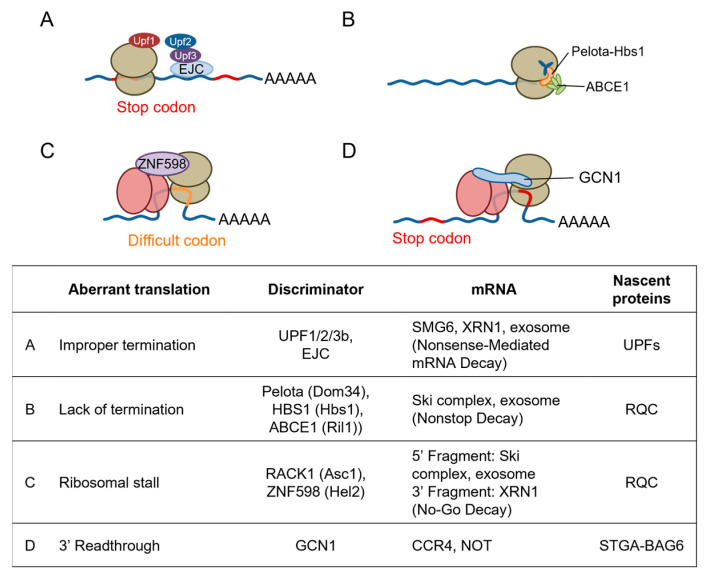
Quality control of aberrant mRNA translation. (**A**) The nonsense-mediated mRNA decay (NMD) pathway recognizes and eliminates mRNAs carrying a termination codon at an aberrant position. NMD proceeds by recruiting the NMD machinery, including up-frame shift proteins, via the exon junction complex, which is located upstream of the exon–exon junction. A premature translation termination codon leads to the dynamic assembly of up-frame shift proteins. (**B**) Nonstop mRNAs lacking a termination codon because of the incorrect attachment of a poly(A) tail to an open reading frame are eliminated by nonstop decay (NSD). The yeast translation factor complex Dom34–elongation factor 1 alpha-like protein Hbs1 (Pelota–Hbs1 in mammals) promotes the dissociation of the translation elongation complex, mRNA, and peptidyl-tRNA. The RNase L inhibitor 1 (RLI1, ATP-binding cassette sub-family E member 1, ABCE1 in mammals) is a ribosomal recirculation factor that cooperates with Dom34–Hbs1 to dissociate ribosomes. (**C**) The disome formed at a difficult codon triggers the recruitment of the zinc finger protein 598 (ZNF598) in mammals (E3 ubiquitin-protein ligase Hel2 in yeast) and the subsequent ubiquitination of 40S ribosomal subunit proteins. (**D**) The readthrough of the stop codon results in the incorporation of amino acids into the nascent polypeptide chain without proper termination. Because most mRNAs have an additional termination codon in the 3′ UTR, the ribosome cannot reach the poly(A) tail and stalls, resulting in the formation of a disome. GCN1 binds to the disome and recruits the CCR4-Not complex 3’-5’-exoribonuclease subunit Ccr4 (CCR4/NOT) complex to degrade the stalled mRNA.

**Figure 5 ijms-25-02998-f005:**
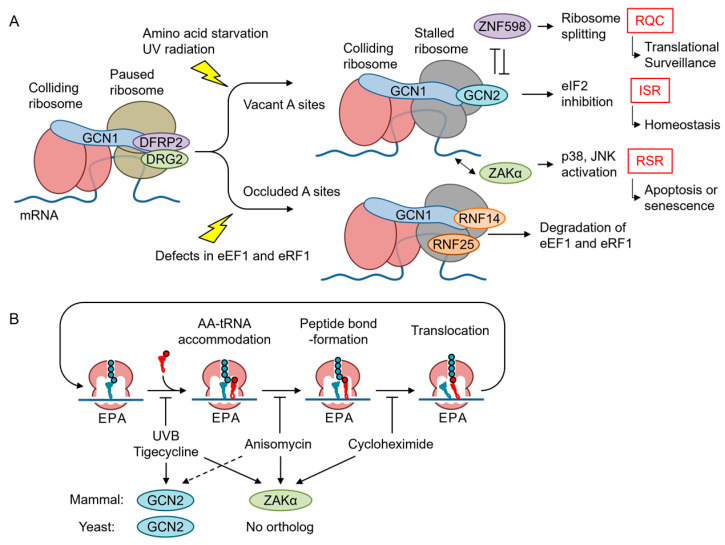
Role of GCN1 in ribosomal stress surveillance. (**A**) Mammalian GCN1 detects ribosome collision by direct binding, as observed in yeast, and is recruited by ZAKα to the disome fraction. GCN1 can recruit the downstream effectors RNF14 and CCR4/NOT to resolve stalled ribosomes by ribosome-associated quality control (RQC), as well as GCN2, to activate ISR and attenuate the global translation initiation depending on stress severity. (**B**) The inhibition of translation elongation can trigger both the GCN2–ISR and the ZAKα–ribotoxic stress response (RSR) pathways. GCN2 is activated in response to UVB irradiation and elongation inhibitors, such as tigecycline, which stall ribosomes with a vacant A-site, although intermediate anisomycin can also activate GCN2 in human MCF10A cells but not in yeast cells (dashed line). ZAKα is activated by a broad spectrum of elongation inhibitors, including cycloheximide, and is conserved in animals but absent in yeast.

**Figure 6 ijms-25-02998-f006:**
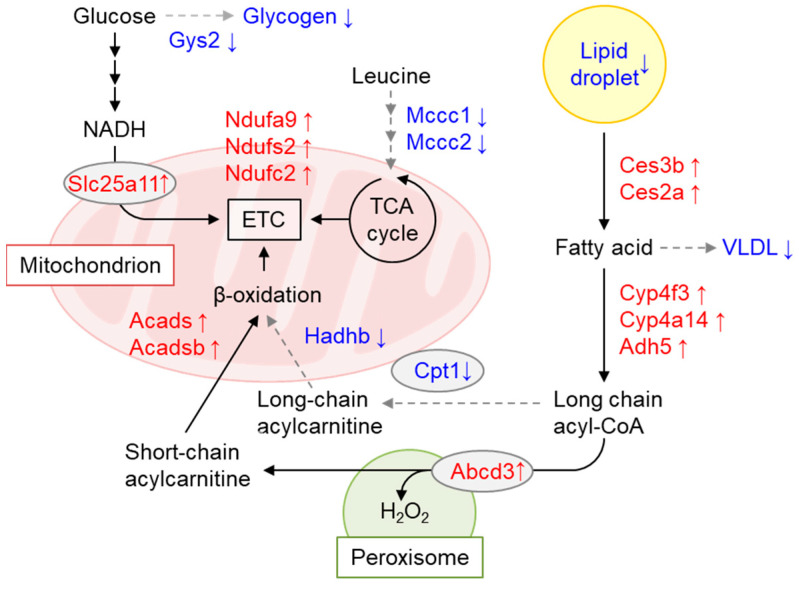
Altered hepatic metabolism in tamoxifen-induced *Gcn1* CKO mice. A combination of tamoxifen administration and *Gcn1* CKO in mice resulted in a decrease in liver and white adipose tissue weight, reduced hepatic lipid storage, as well as reduced glucose and very-low-density lipoprotein levels in the blood. The alteration of the liver proteome in CKO mice implicates a decrease in glycogen deposition, leucine degradation, and fatty acid oxidation (indicated in blue), and an inverse increase in the peroxisomal pathway, resulting in inefficient energy production (indicated in red) [127]. Solid arrows indicate the probable increment in the metabolic flux and gray dashed arrows indicate the probable decrease in the metabolic flux, respectively. Triple arrows (→→→) represent the multiple steps of the metabolisms.

**Table 1 ijms-25-02998-t001:** Molecular properties of RWD-domain-containing proteins.

RWD-Domain-Containing Proteins	Gene Symbol	Amino Acids	RWD Domain Region
eIF-2-alpha kinase GCN2	EIF2AK4 (GCN2)	1649	25–137
Protein IMPACT	IMPACT	320	14–116
Ring finger protein 14	RNF14	474	11–137
Ring finger protein 25	RNF25	459	18–128
RWD domain-containing protein 1	RWDD1 (DFRP2)	243	10–114
RWD domain-containing protein 2A	RWDD2A	292	14–134
RWD domain-containing protein 2B	RWDD2B	319	41–165
RWD domain-containing protein 3	RWDD3 (RESUME)	267	7–114
RWD domain-containing protein 4	RWDD4	188	9–111
WD repeat-containing protein 59	WDR59	974	393–494

## Data Availability

Not applicable.

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
