# Peer review of "Emerging Role of GCN1 in Disease and Homeostasis"

_ijms, 2024, doi:10.3390/ijms25052998_

Round 1
Reviewer 1 Report
Comments and Suggestions for Authors
The review by Tatara et al. focuses on the functions of the GCN1 protein in activating the amino acid starvation sensor GCN2. In addition to this well-known role, the review presents GCN1 as a multifunctional protein that is also involved in a number of other cellular processes. For example, recent studies using genome editing technologies and cryo-electron microscopy have shown that GCN1 is involved in overcoming ribosome stalling and regulating energy metabolism at the whole organism level. The authors make the logical assumption that GCN1 exerts its functions through an unexplored mechanism of switching its protein partners, proteins containing RWD domains.
The review carefully and thoroughly describes the canonical function of GCN1 as an activator of the amino acid starvation sensor, with special attention to the mechanisms of activation of the amino acid deficiency response. The authors not only list the sequence of events that occur during translation regulation as a result of GCN2 activation, but also cover in detail all known mechanisms of cell response to such stress. In addition, the mechanisms of co-translational quality control, ribosome release from damaged matrix, and ribosome stalling are discussed in detail.
The review is written in good scientific language and contains a comprehensive description of the molecular mechanisms mentioned. The manuscript is well illustrated, the schemes presented by the authors are logical, easy to read and complement the text perfectly. From my point of view, the authors have managed to clarify a rather complex topic in an orderly and complete manner, not only listing the facts known so far, but also analyzing them and expressing their assumptions about the functions of GCN1. The review is fully in line with the direction of the journal. After reading the submitted manuscript, I enjoyed and learned a lot. I do not have any significant remarks or negative comments. I believe that the review by Tatara et al. can be published as submitted.
Author Response
Response: Thank you very much for accepting our manuscript. We are very encouraged by your comments. We would like to proceed as it is.
Reviewer 2 Report
Comments and Suggestions for Authors
Tatara et al., summarize roles of GCN1 very well in this review. Where appropriate, the authors contrast reported findings to GCN2. Attention is paid to non-canonical functions and GCN1's role in aging, neurodegeneration, and other disease states.
The review is well-written, concise but substantial, and a great resource for the whole field. Minor suggestions are:
Due to the number of abbreviations used throughout the text, it might help to provide a table or a designated segment to explain them. As abbreviations cannot be avoided for protein names, I would suggest using the full name for everything else (for example, halifuginone instead of HF). Protein names should be consistently spelled out at first mention.
Line 122 a space is missing.
Line 123 has an extra space after TNFa.
Figure 3D is very difficult to read - color coding amino acids could help generally increasing the size.
Line 572: Is MISTERMINATE an acronym?
Author Response
Tatara et al., summarize roles of GCN1 very well in this review. Where appropriate, the authors contrast reported findings to GCN2. Attention is paid to non-canonical functions and GCN1's role in aging, neurodegeneration, and other disease states.
Response: We appreciate your careful review of our manuscript. We have addressed your concerns to the best of our ability and have highlighted the revised parts of the manuscript in yellow.
The review is well-written, concise but substantial, and a great resource for the whole field. Minor suggestions are:
Due to the number of abbreviations used throughout the text, it might help to provide a table or a designated segment to explain them. As abbreviations cannot be avoided for protein names, I would suggest using the full name for everything else (for example, halifuginone instead of HF). Protein names should be consistently spelled out at first mention.
Response: We have revised the manuscript to avoid abbreviations and use full names whenever possible. Protein names were corrected to be spelled consistently in the first mention. A new list of abbreviations has been added with a section at the end of the manuscript.
Line 122 a space is missing.
Response: The missing space in line 129 has been inserted.
Line 123 has an extra space after TNFa.
Response: The extra space on line 130 has been removed.
Figure 3D is very difficult to read - color coding amino acids could help generally increasing the size.
Response: Sorry for the figure is not clear. Figures 3D have been replaced with a clearer image.
Line 572: Is MISTERMINATE an acronym?
Response: Mitochondrial-stress-induced translational termination impairment and protein carboxyl terminal extension is termed MISTERMINATE. MISTERMINATE is partially acronym but is a name, so we have used it as is in line 597.
Reviewer 3 Report
Comments and Suggestions for Authors
The review article by Tatara et al. examined the evolving role of GCN1 in disease and homeostasis, emphasizing its canonical involvement in amino acid response and novel functions in cell processes like proliferation and apoptosis. I have several comments which may improve the manuscript:
1. Consider making a color adjustment for gene RNF14 in Figure 2. The yellow is not reader-friendly.
2. For Figures 3A–C,. The confidence of the predicted model is colored blue, cyan, yellow, and orange. However, I cannot find protein domains in yellow or orange. For Figure 3D-E, the texts are quite blurry, making it difficult for the reader to read comfortably. Please use a vector figure here instead.
3. In Figure 6, the triple arrows (->->->) are unclear in meaning and the gray rectangle should not contain arrows (e.g. over the text "Long-chain acylcarnitine”). Please make the figure legend consistent.
Author Response
The review article by Tatara et al. examined the evolving role of GCN1 in disease and homeostasis, emphasizing its canonical involvement in amino acid response and novel functions in cell processes like proliferation and apoptosis. I have several comments which may improve the manuscript:
Response: We appreciate your careful review of our manuscript. We have addressed your concerns to the best of our ability and have highlighted the revised parts of the manuscript in yellow.
- Consider making a color adjustment for gene RNF14 in Figure 2. The yellow is not reader-friendly.
Response: Sorry for being unclear. The color of RNF14 in Figure 2 was changed.
- For Figures 3A–C,. The confidence of the predicted model is colored blue, cyan, yellow, and orange. However, I cannot find protein domains in yellow or orange. For Figure 3D-E, the texts are quite blurry, making it difficult for the reader to read comfortably. Please use a vector figure here instead.
Response: Figures 3A and B have been replaced with clearer images. The orange color, which indicates very low prediction accuracy, is not included in the models, and this has been noted in the legend. Figures 3D and E have been replaced with vector images.
- In Figure 6, the triple arrows (->->->) are unclear in meaning and the gray rectangle should not contain arrows (e.g. over the text "Long-chain acylcarnitine”). Please make the figure legend consistent.
Response: Explanation for arrows was added in the figure legend. The background color of mitochondrion was adjusted and gray rectangles were removed from the figure.